



# Ionospheric Response to Solar EUV Radiation Variations: Comparison based on CTIPe Model Simulations and Satellite Measurements

Rajesh Vaishnav[1], Erik Schmölter[2], Christoph Jacobi[1], Jens Berdermann[2], and Mihail Codrescu[3]

[1]Leipzig Institute for Meteorology, Universität Leipzig, Stephanstr. 3, 04103 Leipzig, Germany
[2]German Aerospace Center, Kalkhorstweg 53, 17235 Neustrelitz, Germany
[3]Space Weather Prediction Centre, National Oceanic and Atmospheric Administration, Boulder, Colorado, USA

**Correspondence:** Rajesh Vaishnav (rajesh_ishwardas.vaishnav@uni-leipzig.de)

**Abstract.** The ionospheric Total Electron Content (TEC) provided by the International GNSS Service (IGS), and the Coupled Thermosphere Ionosphere Plasmasphere Electrodynamics (CTIPe) model simulated TEC have been used to investigate the delayed ionospheric response against solar flux and its trend during the years 2011 to 2013. The analysis of the distinct low and mid-latitudes TEC response over 15°E shows a better correlation of observed TEC and the solar radio flux index F10.7 in

the Southern Hemisphere compared to the Northern Hemisphere. Thus, a significant hemispheric asymmetry is observed.

The ionospheric delay estimated using model simulated TEC is in good agreement with the delay estimated for observed TEC against Solar Dynamics Observatory (SDO) EUV Variability Experiment (EVE) measured flux. The average delay for the observed (modeled) TEC is 17(16) h. The average delay calculated for observed and modeled TEC is 1 and 2 h longer in the Southern Hemisphere compared to the Northern Hemisphere.

Furthermore, the observed TEC is compared with the modeled TEC simulated using the SOLAR2000 and EUVAC flux models within CTIPe over Northern and Southern Hemispheric grid points. The analysis suggests that TEC simulated using the SOLAR2000 flux model overestimates the observed TEC, which is not the case when using the EUVAC flux model.

## 1 Introduction

The ionospheric day to day variations are mainly controlled by fluctuations of solar extreme ultraviolet/ultraviolet (EUV/UV)

radiation responsible for photoionization and photo-dissociation processes, lower atmospheric forcing, and space weather events such as geomagnetic storms. During geomagnetically and meteorologically quiet conditions, the electron density gradually increases after sunrise with maximum around 14:00 LT due to the photo-chemical processes, and starts decreasing thereafter due to the combined effect of production and strong recombination, continuing after sunset due to recombination processes.

The solar radiation flux varies at different time scales, including the diurnal cycle, the 27-day solar rotation period, and the prominent 11-year solar cycle. This results in corresponding variations in composition and dynamics of the thermosphere-





ionosphere (T/I) system (Hedin, 1984). The T/I system is highly variable with location and time, depending on the solar activity and geomagnetic disturbances.

The photoionization processes in the ionosphere cause different variations, including short term variability at the time scale of the 27-day solar rotation or seasonal variations. Past studies on the effect of solar radiation variations at different timescales have been based on the total electron content (TEC, frequently given in TECU, 1 TECU=$10^{16}$ electrons $m^{-2}$), peak electron density (NmF2, $cm^{-3}$), and the corresponding height (HmF2, km) (e.g., Jakowski et al., 1991; Afraimovich et al., 2008; Lee et al., 2012; Jacobi et al., 2016; Schmölter et al., 2018, 2020; Vaishnav et al., 2018, 2019; Ren et al., 2018, & references therein).

The annual contributions to the mean TEC variability has a stronger impact on the Southern Hemisphere, wheras the semi-annual contribution have similar phase and amplitude at conjugate points suggesting close coupling between the ionosphere and thermosphere (Liu et al., 2009). Mendillo et al. (2002) suggested that both annual and semiannual variations of NmF2 are largely caused by changes in the neutral composition, which are driven by the global thermospheric circulation.

Solar proxies are frequently used to represent the solar activity. Among them are the F10.7 index, the Mg-II index, and the He-II index. Furthermore, attempts have been made to determine simple proxies for global TEC variability based on these indices (e.g., Unglaub et al., 2011). These proxies have been compared to the ionospheric parameters at the time scale of the 27-day solar rotation. An ionospheric delay about 1-2 d have been reported (e.g., Jakowski et al., 1991; Jacobi et al., 2016). Using more precise and high temporal resolution solar flux, an ionospheric delay of about 17-19 h has been reported by Schmölter et al. (2018). The spatial and seasonal effects on the ionospheric delay have been further investigated in detail by Schmölter et al. (2020) using European and Australian locations. Their study highlighted the role of geomagnetic activity on the ionospheric delay.

To investigate the process associated with the ionospheric delay, Jakowski et al. (1991) used a one-dimensional numerical model between 100 to 250 km altitude with simplifying assumptions. They suggested that a delay of approximately 2 d arises in atomic oxygen at 180 km due to photo-dissociation and transport processes. This hypothesis has yet to be confirmed with comprehensive ionospheric models such as CTIPe. Ren et al. (2018) investigated the ionospheric delay using observations and modeling. They emphasized the role of the $[O]/[N_2]$ ratio in the ionospheric delay. Vaishnav et al. (2018) suggested a possible role of transport processes in the ionospheric delay.

During the past decades more improved physics-based T/I models have been developed, which are able to characterize ionospheric dynamics. Among them are the Coupled Thermosphere Ionosphere Plasmasphere Electrodynamics (CTIPe, Fuller-Rowell and Rees, 1983; Codrescu et al., 2012), the Thermosphere-Ionosphere- Electrodynamics General Circulation Model (TIE-GCM, Richmond et al., 1992) and the Global Ionosphere Thermosphere Model (GITM, Ridley et al., 2006). Furthermore, some extended Earth system models like WACCM-X (Liu et al., 2018) and GAIA (Jin et al., 2012; Liu et al., 2020) include T/I dynamics. Based on the results of the T/I model, an ionospheric lag against variations of the solar EUV could be identified, whereby the EUV entry in the model was represented by the F10.7 index (Ren et al., 2018; Vaishnav et al., 2018).

The most commonly used solar proxy for ionizing irradiance is the solar radio flux at 10.7 cm (F10.7 index, given in solar flux units (sfu), 1 sfu =$10^{-22} W m^{-2} Hz^{-1}$) (Tapping, 1987). Most of the T/I models use a modified F10.7 index (e.g., the average



of daily and 41 or 81 days averages) to calculate the model EUV spectra based on reference spectra. Several authors have reported that a modified F10.7 index, which includes both short term and long term variability, is a better proxy for ionizing irradiance than F10.7 directly (Richards et al., 1994). There are several empirical models available, such as the SOLAR2000

(Tobiska et al., 2000) and EUVAC flux model (Richards et al., 1994), to calculate the irradiance.

Profiles of the delayed ionospheric response dependent on latitude have been calculated in previous studies (Lee et al., 2012; Ren et al., 2018) and the influence of seasonal variations and geomagnetic activity on both hemispheres has also been characterized (Schmölter et al., 2020). The complexity of the seasonal variations and associated anomalies has been investigated in other studies for ionospheric parameters like TEC (Romero-Hernandez et al., 2018). Such seasonal anomalies were observed

in the F2 region associated with higher electron density in winter than in summer during daytime (the so-called winter or seasonal anomaly), during equinoxes than during solstices (semiannual anomaly), and in December than in June (annual or non seasonal anomaly) (Balan et al., 1998; Zou et al., 2000; Romero-Hernandez et al., 2018). However, seasonal variations have not yet been analyzed for the ionospheric delay.

The ionospheric composition is mainly controlled by the photoionization, the loss through recombination, and transport

processes. Transport processes play a significant role in the T/I composition and are responsible for the plasma distribution possibly leading to the observed ionospheric anomalies. Fuller-Rowell (1998) suggests a possible mechanism associated with the seasonal anomaly through the neutral wind.

This study aims to analyze the ionospheric TEC variations in both the Northern and Southern Hemisphere during a moderate solar activity phase within solar cycle 24, (2011-2013). We use GNSS data from $70°S$ to $70°N$ latitude at $15°E$ longitude due

to better coverage with ground measurements in TEC maps. The observed TEC is compared with the model simulated TEC using different solar EUV flux models. The ionospheric delay against solar EUV flux has been investigated by Schmölter et al. (2020) using TEC observations. Therefore, the focus of the present study is laid on the ability to reproduce the ionospheric delay using the CTIPe model at $15°E$.

In Section 2, we introduce the data sources and the CTIPe model. In Section 3, we investigate the TEC variability, a possible

relationship with F10.7 index variations, and compare TEC simulated with the different solar EUV flux models. In Section 4, we summarize our conclusions.

## 2   Observations and Model

### 2.1   TEC observations

In this paper, we use TEC data from $70°S$ to $70°N$ latitude at $15°E$ from the International GNSS Service (IGS) provided by

NASA's CDDIS (Noll, 2010), which are available at 1 h time resolution and with a latitude-longitude resolution of $2.5° \times 5°$ (Hernández-Pajares et al., 2009).





## 2.2 Solar EUV radiation

Several solar proxies are available that have frequently been used in previous studies to represent the solar activity level compared to the ionospheric parameters before the space age, and due to the unavailability of direct solar EUV measurements.

Continuous time series of the solar EUV spectrum itself, however, are available since the launch of the NASA Thermosphere ionosphere Mesosphere Energetics and Dynamics (TIMED) satellite mission in 2001. Solar irradiance measurements from the TIMED Solar Extreme Ultraviolet Experiment (SEE) instrument are available since 22 January 2002 (Woods et al., 2005). The SEE instrument is designed to measure the soft X-rays and EUV radiation from 0.1 to 194 nm with resolution and accuracy of 0.1 nm and approximately 10-20 %. SEE includes two instruments, the EUV grating spectrograph and the XUV photometer

system (Woods et al., 2000). Here we use daily values of solar irradiance integrated from 1 to 105 nm wavelength. The TIMED SEE observations are used for comparison with the empirical solar flux models, SOLAR2000 and EUVAC.

Furthermore, the Solar Dynamics Observatory (SDO) EUV Variability Experiment (EVE) provides a continuous high-resolution spectrum with a wavelength range from 0.1 to 120 nm, a spectral resolution of 0.1 nm, and a temporal resolution of 20 s. (Woods et al., 2010; Pesnell et al., 2011). The high resolution EUV observations provided by SDO EVE satellite have

been used to calculate an ionosphere delay in TEC.

Solar proxies are mostly used as a solar activity representation in thermosphere-ionosphere models. Hence, we also use the daily F10.7 index for our analysis.

## 2.3 CTIPe Model

The CTIPe model is a global, first principle, three dimensional numerical, physics-based coupled thermosphere-ionosphere-

plasmasphere model, which self-consistently solves the primitive equations of continuity, momentum, and energy to calculate wind components, global temperature, and the composition of neutrals, which is further used to calculate plasma production, loss, and transport. The model consist of four components, namely a neutral thermosphere model (Fuller-Rowell and Rees, 1980), a mid- and high-latitude ionosphere convection model (Quegan et al., 1982), a plasmasphere and low latitude ionosphere model (Millward et al., 1996), and an electrodynamics model (Richmond et al., 1992). The calculations are performed with

2°/18° latitude/longitude resolution. In the vertical direction, the atmosphere is divided into 15 levels in logarithmic pressure at an interval of one scale height, starting from a lower boundary at 1 Pa (about 80 km altitude) to above 500 km altitude at pressure level 15. The high latitude ionosphere (above 55° N/S) and the mid-low latitude ionosphere-plasmasphere components have been implemented as separate modules.

The numerical solution of the composition equation and the energy and momentum equations describe transport, turbulence,

and diffusion of atomic oxygen, molecular oxygen, and nitrogen (Fuller-Rowell and Rees, 1983). To run the model, external inputs are required like solar UV and EUV, Weimer electric field, TIROS/NOAA auroral precipitation, and tidal forcing from the Whole Atmosphere Model (WAM). The F10.7 index is used as an input solar proxy to calculate ionization, heating, and oxygen dissociation processes in the ionosphere. The CTIPe ionosphere results include the major ion species $H^+$ and $O^+$ for





all altitudes, and other molecular and atomic ions $N_2$, $O_2$, $NO^+$ and $N^+$ below 400 km. Detailed information on the CTIPe

model is available in Codrescu et al. (2008, 2012); Fernandez-Gomez et al. (2019).

## 2.4 EUV representation in the CTIPe model

### 2.4.1 SOLAR2000 model

The SOLAR2000 model is the most recent EUV model version in a series of iterations by Tobiska et al. (2000). SOLAR2000 incorporates multiple satellite and rocket measurements, including the AE-E satellite observations, to specify both a reference

spectrum and solar variability. The EUV is calculated using the Lyman $\alpha$ flux and the F10.7 index with the set of modeling equations. SOLAR2000 determine the EUV irradiance for 809 emission lines and also for 39 wavelength bands.

### 2.4.2 EUVAC solar flux model

Within CTIPe, a reference solar spectrum based on the EUVAC model (Richards et al., 1994) and the Woods and Rottman (2002) model, driven by variations of input F10.7 is used. The EUVAC model is used between 5 nm and 105 nm, and the

Woods and Rottman (2002) model for 105 nm to 175 nm. Solar flux is obtained from the reference spectra using the following equation:

$$f(\lambda) = f_{ref}(\lambda)[1 + A(\lambda)(P - 80)] \tag{1}$$

where $f_{ref}$ and A are the reference spectrum and solar variability factor, respectively, and $P = 0.5 \times (F10.7 + F10.7A)$, where F10.7A is the average of F10.7 over 81 days.

The EUVAC model includes solar flux in 37 wavelength bins based on the measured F74113 solar EUV reference spectrum (Hinteregger et al., 1981) and the solar cycle variation of the flux.

### 2.4.3 Comparisons between empirical EUV irradiance variability models and observations

We compare TIMED SEE observations with the two empirical models constructed from direct proxy parameterizations of the EUV irradiance data base, which are used to represent EUV in the CTIPe model.

Figure 1 shows the modeled integrated irradiance spectra from 5 to 105 nm calculated by both models together with the TIMED SEE irradiance from 2011 to 2013. The second y-axis shows the F10.7 index used to calculate the spectra in empirical models. In comparison to the SOLAR2000 model flux and TIMED SEE, flux values calculated by the EUVAC model are smaller. There is a significant difference between the flux models and the observed irradiance. The flux calculated by the SOLAR2000 model overestimates the observed flux mostly during the Northern Hemisphere winter months, whereas it is in

good agreement during Northern Hemisphere summer months. The observed EUV irradiance during moderate solar activity is comparable to the SOLAR2000 flux, with a difference of about 10% and 23% higher than the EUVAC model. The EUVAC flux is about 30% lower than the SOLAR2000 model. The correlation coefficient of EUV from both the EUV flux models with



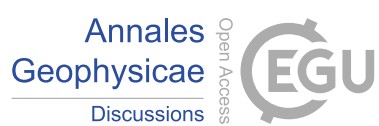

the observed EUV flux is approximately 0.90 during the study period. In summary, the SOLAR2000 model is in relatively good agreement with the observed flux while the EUVAC model underestimates SOLAR2000 and the TIMED SEE flux. These results agree with earlier comparisons (Lean et al., 2003; Woods et al., 2005; Lean et al., 2011, & references therein).

Woods et al. (2005) compared the TIMED SEE observations with the flux calculated from different empirical models for 8 February 2002. They reported that the empirical models are within 40% of the SEE measurement at wavelengths above 30 nm. The EUVAC and SOLAR2000 models agreed best with TIMED SEE, compared to the other models.

Lean et al. (2003) validated the NRLEUV model with different empirical models such as SOLAR2000, HSG, and EUVAC. In absolute scales NRLEUV, HFG and EUVAC have total energies that agree within 15%, but the SOLAR2000 absolute scale is more than 50% higher. Their study reveals that long EUV wavelength (70-105 nm) energy contributions (about 46% of the whole flux from 5 to 105 nm) is the main reason for higher EUV flux in the SOLAR2000 model compared to other empirical models.

## 3 Results and discussion

In the following sections, we show the results and discuss the TEC observations and their comparison with the modeled TEC at $15°$E. Furthermore, relations with solar radiation and the delayed response over both the Northern and Southern Hemispheres are presented. Schmölter et al. (2020) has reported on a detailed investigation of the delayed ionospheric response over European and Australian regions. Here, we analyze the delayed response at $15°$E covering the latitudes from $70°$S to $70°$N and compare the response over the South African region with the European region.

In this study we have addressed the following points:

1. The TEC variations at moderate solar activity of solar cycle 24 are analyzed to compare the input for the delay analysis. A characterization of these differences between observed and modeled TEC is important to derive further relations.

2. We used the periodicity estimation (frequency analysis) to study observed and modeled TEC characteristics in detail.

3. The relation between the F10.7 index and hemispheric TEC has been used to analyze the solar and ionospheric inputs of delay estimation.

4. In our study we focus on the ionospheric delay estimation as a main point of our analysis.

5. Observed TEC variations and its comparison with simulated TEC is done by using different flux models. In previous work it has already be shown that the solar activity has the strongest impact on TEC under nominal conditions and is therefore significant for the derived delay.

### 3.1 TEC variation at moderate solar activity of solar cycle 24

The ionospheric electron density is strongly varying from day to night depending on the daily variations of solar radiation.





Figure 2 depicts the mid-day (11:00-13:00 LT) variations in TEC for the moderate solar activity conditions from 2011 to 2013. The Figure shows the comparison between the observed TEC and modeled TEC simulated using the EUVAC flux model at $15°$ E longitude. Note that at this longitude, climatological hemispheric differences in TEC are expected due to peculiarities
of the magnetic field, in particular the South Atlantic Anomaly, which causes low ionization in the Southern Hemisphere.

The TEC variations highly depend on the level of ionization due to the solar radiation flux. The observed TEC shows such variations compared to the SDO EVE integrated flux (1-120 nm), as shown on the second y-axis of Figure 2. During 2012, there are continuous 27 days cycles. This kind of regular variations in solar observations enables us to explore the respective ionospheric variations, which are clearly driven by the ionization and recombination processes.

The maximum TEC is observed at the equator and in low latitude regions. The TEC level reduces towards the high latitude regions. In general, the TEC values varying latitudinally depending on the Northern and Southern Hemispheric season. At the equator, the plasma moves upward and redistributes along the equator, causing the Fountain effect (Appleton, 1946). The thermospheric wind circulations firmly controls the plasma movement. The plasma moves from the summer hemisphere to the winter one, causing a decrease in the F peak height, further decreasing the $O/N_2$ ratio. The TEC values in the Southern
Hemisphere are higher than in the Northern Hemisphere.

The Figure 2(a) shows the two peaks of ionization during the spring 2011, but in autumn the maximum is shifted towards winter, clearly solar driven, and in 2013 there are local minima during equinox.

In comparison to observed TEC, the modeled TEC (Figure 2(b)) is lower during the spring and summer period in the Southern Hemisphere, while it is in better agreement during the winter season. The bias between the modeled and observed
TEC is higher during the spring and summer season. In general, the modeled TEC is lower than the observed TEC.

The variations in TEC are not only controlled by the solar radiation, but there are other factors such as local dynamics or geomagnetic activities due to solar wind variations, which also influence the ionospheric state (Abdu, 2016). Fang et al. (2018) studied day to day ionospheric variability and suggested that absolute values in TEC variability at low latitudes are largely controlled by solar activity, while for mid- and high-latitudes, however, solar and geomagnetic activities contribute roughly
equally to the absolute TEC variability.

A detailed comparison between the observed TEC and modeled TEC simulated using the different solar flux models (SO-LAR2000 and EUVAC) during January, June, and December is presented and discussed in Section 3.6.

## 3.2 Periodicity estimation

Solar activity varies at different time scales from minutes to years or even centuries. The periodic behavior in the solar proxies
has been studied by various authors to explore the response of the terrestrial atmosphere and especially the T/I region, and to investigate the connection between solar variability and ionospheric parameters (Jacobi et al., 2016; Vaishnav et al., 2019). A widely used method to analyse periodicities in time series is the continuous wavelet transform (CWT). The CWT captures the impulsive events when they occur in the time series (Percival and Walden, 2000; Mallat, 2009). However, the CWT also reveals lower frequency features of the data hidden in the time series.



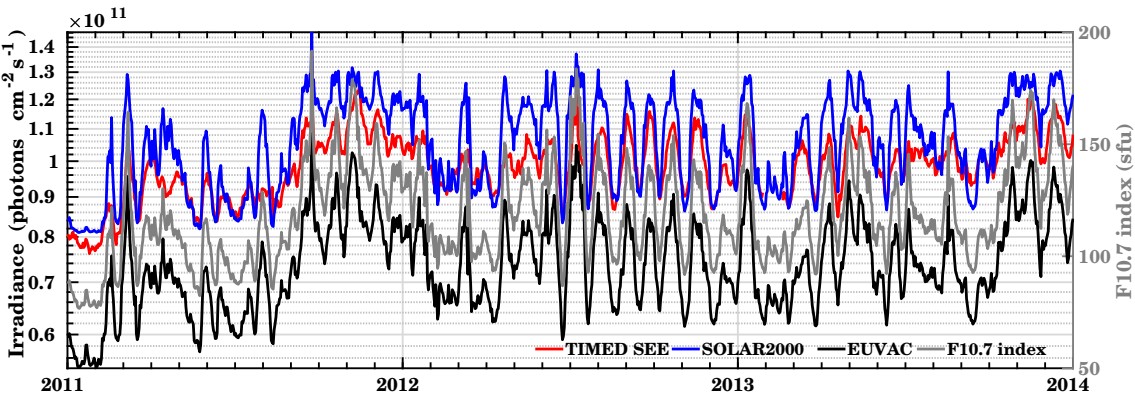

**Figure 1.** Time series of 0.5 to 105 nm integrated daily irradiance from 2011 to 2013 estimated from TIMED SEE observations, SOLAR2000, and EUVAC. The right y-axis represents the F10.7 index.

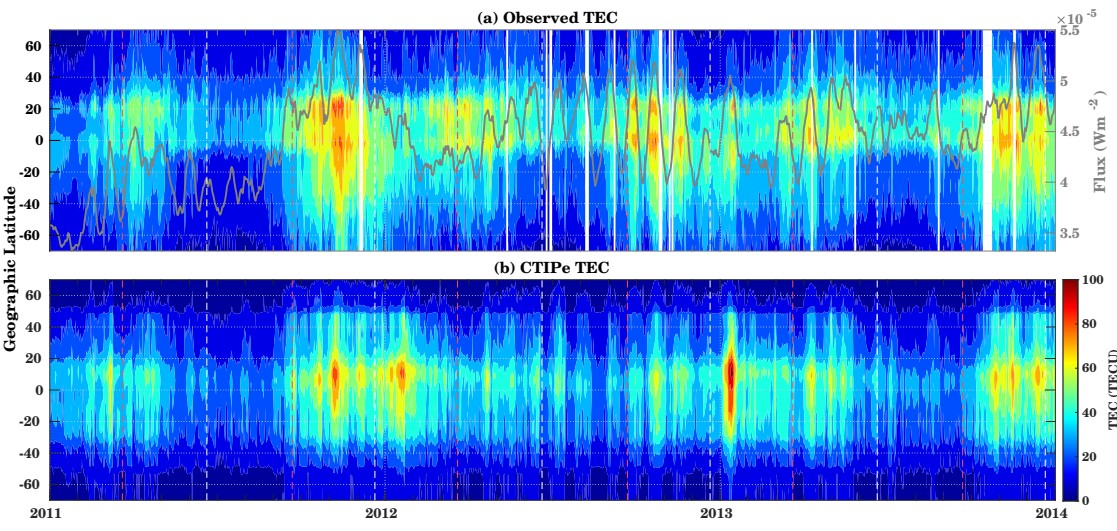

**Figure 2.** Latitudinal variation of (a) observed TEC and (b) model simulated TEC around noon (11:00-13:00 LT) at $15°$E longitude. The gray curve in panel (a) represents the SDO EVE integrated flux (1-120 nm).





Here, we will investigate and compare the different temporal patterns of observed and modeled TEC. The daily TEC and F10.7 index from 2011 to 2013 are used to analyze the periodic behavior of the T/I system. Figure 3 shows the continuous wavelet spectra of the model simulated TEC, observed TEC, and F10.7 for low [$\pm30°$], mid [$\pm(30°-60°)$], and high [$\pm(60°-70°)$]] latitudes from 2011 to 2013.

The upper panels (a-c) of Figure 3 show the CWT of modeled TEC while the middle panels (d-f) show the observed TEC,

respectively, over the different latitude bands mentioned in the Figure title. The lower panel shows the CWT of F10.7.

The CWT of modeled TEC shows the dominant 16-32 d oscillations during 2012. A similar 16-32 days periodicity is observed in the F10.7 index. It is well known that the 27 d periodicity is one of the major and dominant modes of variations in the solar proxies. This is, however, not the case during 2011 and 2013. During these periods, the influence of other dynamical processes in the ionosphere (e.g., lower atmospheric forcing) is stronger. During these years, very weak 27 d periodicity is

observed. The 27 d period is stronger in the winter season. Pancheva et al. (1991) showed that the 27 d variation in the lower ionosphere (D region) is often caused by dynamical forcing (planetary waves), particularly in the winter season under low solar activity.

As an advantage, the CWT also shows small scale features. Over low- and mid-latitudes, 8-16 d oscillations are observed to be dominant. Furthermore, another high-power region is visible in the 128 to 256 d period, representing the semi-annual

oscillations in both, modeled and observed TEC, and in the F10.7 index. The semi-annual oscillation is mostly dominant during the period of investigation. Apart from it, in model simulated TEC, a 64-128 d period is observed during 2012 and 2013. The oscillations are stronger at low and mid-latitudes stations compared to high latitudes.

The second row of Figure 3 shows the oscillations in the observed TEC. Here, a weak 27 d cycle is observed during December and the 128-256 d period is mostly dominant during 2011 and 2012. There is a weak signature of semi-annual

oscillations during 2013. As compared to the periodicity observed in model simulated TEC, the 64-128 d periodicity is missing in the observations over all the latitudes. Furthermore, shorter-period fluctuations can be seen especially at high latitudes (Figure 3(f)), with a preference for the winter season. These may be connected with planetary wave effects from below (e.g. Altadill et al., 2001, 2003).

Figure 3(g) shows the CWT spectra of the F10.7 index. Here the dominant period is 16-32 d during 2012, and a weak 16-32

d period oscillation is observed during 2011 and 2013.

In general, from the above investigation, it can be seen that 16-32 d periodicity was dominant during 2012. Vaishnav et al. (2019) used cross wavelet and Lomb-Scargle periodogram techniques to estimate the periodicity of various solar proxies and global TEC during long time series from 2000 to 2016. They found that the semi-annual oscillation is mostly dominant during the solar maximum years 2001-2002 and 2011-2012.

## 3.3  Relation between F10.7 index and hemispheric TEC

Solar activity has the strongest effect on ionospheric variations especially during enhanced solar activity. The last solar minimum was extremely extended, and the following solar cycle was quite weak (e.g., Huang et al., 2016), so that meteorological influences become more relevant. To examine the effect of solar activity on TEC variations during a weak solar cycle, we





analysed the relationship between F10.7 and mid-day TEC (11:00-13:00 LT). Figure 4 shows the correlation between TEC
and F10.7 during 2011 to 2013 for the Northern Hemisphere (NH, upper panel) and Southern Hemisphere (SH, lower panel),
indicating the correlation coefficient (R). In order to represent the NH and SH, daily data of $40°$N and $40°$S latitudes at $15°$E
longitude have been used respectively.

We have calculated correlations using the observed TEC over the NH and SH. During 2011, the maximum correlation for
all the years is observed, which amounts to R=0.71/0.79 for the NH/SH. This suggests that mid-day TEC values are mainly
controlled by solar radiation.

From the current study and past publications (Romero-Hernandez et al., 2018), it is well known that during high solar
activity, weak correlations are observed compared to the moderate solar activity conditions. But during the year 2012, the
lowest correlation of about 0.06 was observed in the SH, while the correlation was about 0.36 in the NH region. During the
year 2013, the correlation is weaker than during 2011, namely about 0.42 for the NH and 0.60 for the SH.

In general, the correlation coefficient is higher in the Southern Hemispheric region as compared to the Northern Hemisphere
during 2011 and 2013, whereas lower correlations are observed during the year 2012. The analysis for 2012 shows some
unexpected behavior over these study regions. This unusual behaviour could be due to physical and chemical processes that
have an impact on the ionospheric state, but also the underlying model in the TEC maps is a possible reason for the difference
as there are less ground station in Southern Hemisphere, causing more weight to the model and probably better correlation
with F10.7.

### 3.4   Cross correlation and delay estimation

The possible relations between solar activity, geomagnetic activity, and ionospheric parameters have been studied by several
authors (e.g., Abdu, 2016; Fang et al., 2018; Vaishnav et al., 2019). However, in the past studies, due to the unavailability of
high-resolution data sets, several studies used only daily resolution. To estimate the ionospheric delay, different ionospheric
parameters have been considered using daily resolution data, an ionospheric delay of about 1-2 d against solar proxies has been
reported (Jakowski et al., 1991; Jacobi et al., 2016; Vaishnav et al., 2019). Only recently, Schmölter et al. (2020) used SDO
EVE and GOES EUV fluxes to calculate the ionospheric delay of about 17 h as a mean value based on hourly time resolution
data. This observed delay was also confirmed by numerical physics based models (Ren et al., 2018; Vaishnav et al., 2018).

Here, we investigate the ionospheric delay using hourly resolution observations and compare it with the model simulated
TEC. Figure 5 shows the cross-correlation and a corresponding ionospheric delay calculated using SDO EVE observed inte-
grated flux from 1 to 120 nm wavelength region in comparison with modeled TEC at $15°$E longitude. The modeled TEC used
for these analyses has been simulated using the EUVAC solar flux model and the F10.7 index as a solar input proxy to calculate
the input spectra. The cross correlation was applied monthly from 2011 to 2013 as independent data sets. The upper panel
of Figure 5 shows the (a) cross-correlation and (c) the ionospheric delay using the observed TEC. The maximum correlation
is observed during the year 2012 with about 0.5, while in 2011 and 2013 the correlation is weaker. The lowest correlation is
observed during the winter months of 2011-2012. Further, latitudinal variations are also seen in the correlation coefficient.





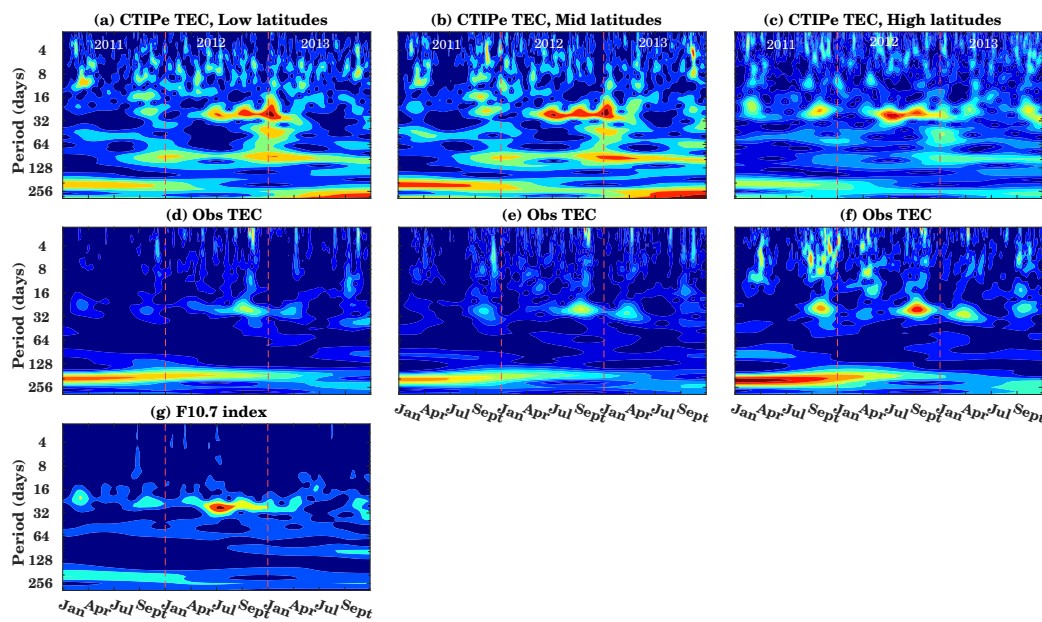

**Figure 3.** Wavelet continuous spectra of daily model TEC (a-c), observed TEC (d-f) for different low [$\pm 30°$], mid [$\pm(30° - 60°)$], and high [$\pm(60° - 70°)$], and (g) F10.7 index.

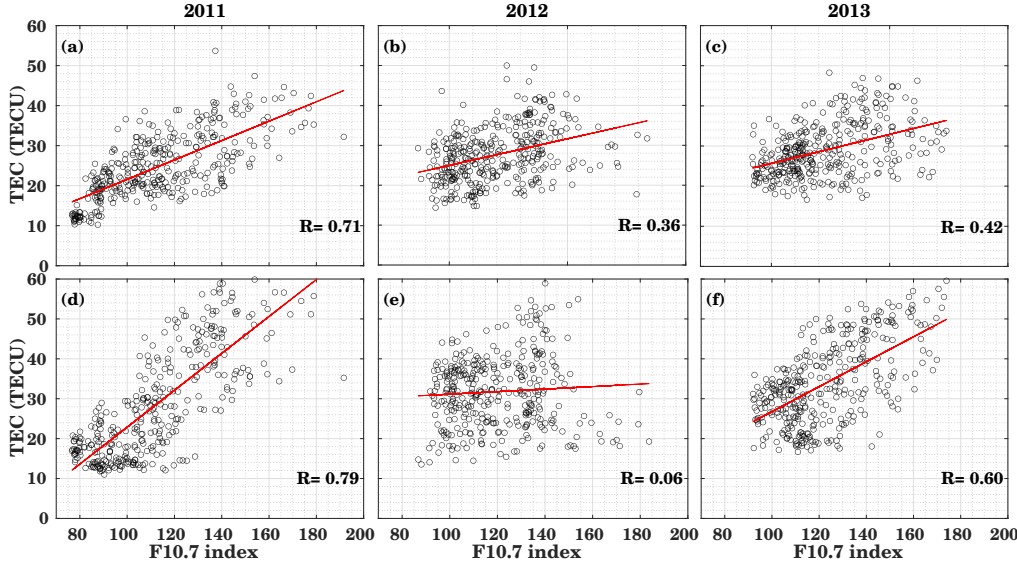

**Figure 4.** Relation between F10.7 index and mid-day observed TEC (11:00-13:00 LT) at $40°$N/$15°E$ (upper panal) and $40°$S/$15°E$ (lower panel) for 2011, 2012 and 2013. The red line is the linear fit.



Figure 5(c) shows the cross-correlation coefficient calculated using the modeled TEC and SDO-EVE flux. The correlation coefficient is higher than the one seen in the observed TEC. In the observed TEC. There are several processes that can influence the behavior of the ionosphere and the real observations such as lower atmospheric forcing or geomagnetic activity. But in the

model, lower atmospheric variability is not included except in a statistical sense, which affects the total variability, hence higher correlation is observed in model TEC compared to observed TEC.

The analysis suggests that the model can reproduce similar trends and features, as shown in the observations. The overall correlation coefficient in the Southern Hemisphere is higher than in the Northern Hemisphere.

Figure 5(b) shows the ionospheric delay calculated from the observed TEC against the SDO flux. The ionospheric delay is

varies strongly with latitude and time. Shorter ionospheric delay is observed during January as compared to other months. For January, the ionospheric delay is about 13-16 h. The maximum delay is about 22 h in the low latitude region during 2011 and 2012, but about 22-23 h during 2013 in low- and mid-latitudes. During 2011 the ionospheric delay is maximum for the winter period at the equator with about 22 h, while it decreases towards high latitudes. A very low ionospheric delay of about 5-10 h is observed during August 2012 for mid-latitudes. As an interesting feature can be noted here that the ionospheric delay is

increasing with increasing solar activity.

A similar analysis for the estimation of the ionospheric delay has been performed for the model simulated TEC, as shown in Figure 5(d). The CTIPe model is able to reproduce features seen in the observed TEC (Figure 5(b)). The ionospheric delay is higher during December and follows the solar activity.

In the higher latitude region (above $60°$ latitude in both hemispheres), the ionospheric delay in the model is smaller than in

the observations and amounts to about 5-10 h. Simultaneously, the correlation coefficient is high at the high latitude regions in the Southern Hemisphere and is about 0.4, as shown in Figure 5(c). This bias is due to the model limitations such as model input, grid resolution and insufficient physical descriptions (Negrea et al., 2012).

Generally, the ionospheric delay calculated from the modeled TEC is in good agreement with the observed one and it is about 17 h. Furthermore, the ionospheric delay is always higher in the Northern Hemisphere as compared to the Southern

Hemisphere. Partly negative correlation has been observed in both the model and the observations. This negative correlation indicates the effect of local dynamics. The correlation coefficients in the Southern Hemisphere are generally higher than in the Northern Hemisphere.

Furthermore, to understand the mean variations of TEC and its connection with the ionospheric delay, we calculated the latitudinal mean observed TEC with the standard deviations and compare it with the model simulated TEC from 2011 to 2013

as shown in Figure 6(a). The observed TEC always overestimated the model simulated TEC at all latitudes. As expected, the maximum TEC of about 50 TECU is observed at low latitudes, while model simulated TEC is about 45 TECU. The maximum bias is observed poleward of $35°$S and $45°$N, and this bias is increasing towards high latitudes. As discussed in the previous sections, there are several problems such as providing inputs for the model, grid resolution effects and insufficient physical descriptions that need to be addressed in the future to reduce the bias in the model.

To see the mean latitudinal variations of ionospheric delay, we used the monthly delay calculated from 2011 to 2013. The mean ionospheric delay is about 17-18 h in the observations at low- and mid-latitudes, while it is about 15 hours in the high



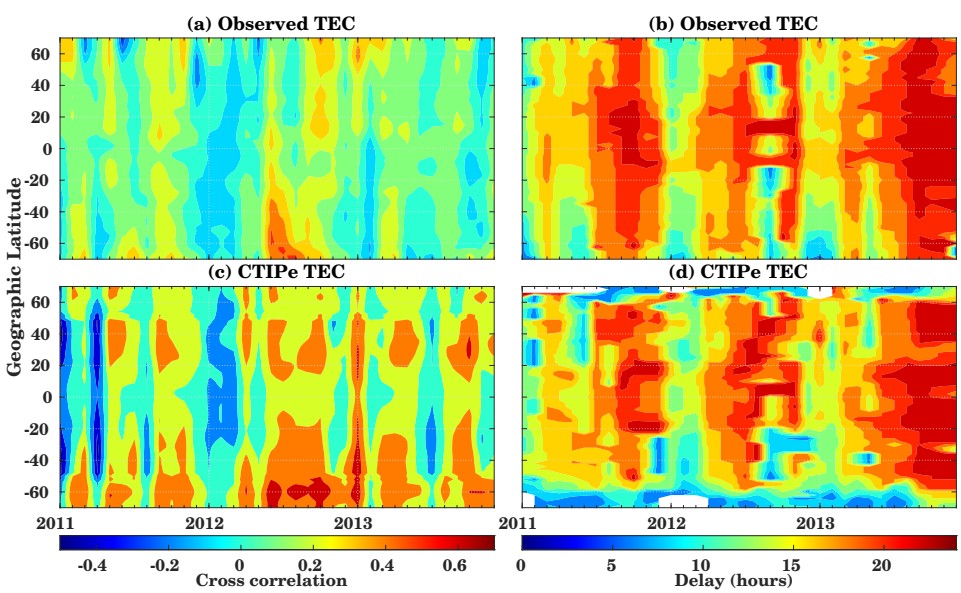

**Figure 5.** Correlation coefficient (a,c, left panels) and delay estimation (b,d, right panels) using observed (upper panels) and model simulated (lower panels) hourly TEC and SDO EVE integrated flux (1-120 nm).

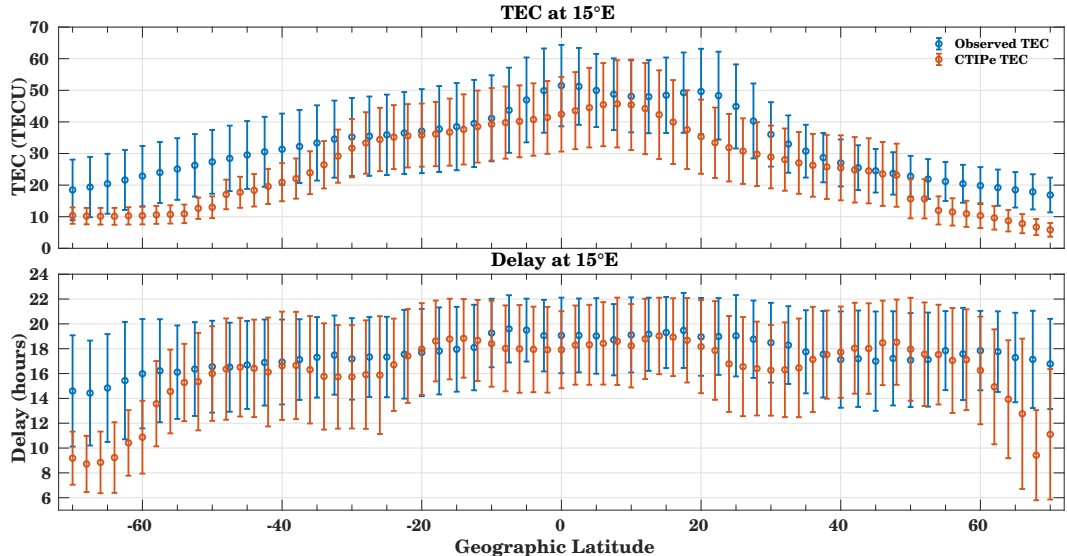

**Figure 6.** (a) Daily mean TEC variations and (b) delay estimation using observed (blue) and model simulated (red) hourly TEC and SDO EVE integrated flux. The error bars show standard deviations of mean values.





latitude regions. As compared to the delay in observations, the model simulated delay is 1-2 h less in the low- and mid-latitudes, but the difference strongly increases in the high latitude regions. Poleward of $55°$, the ionospheric delay reduces to less than 10 h.

This analysis shows that the model can reproduce the ionospheric delay as seen in the observations, and generally produces a delay of about 18 h at middle latitudes.

### 3.5    Observed TEC variations and its comparison to simulated TEC using different EUV flux models

To further visualize the observed daily TEC and its comparison with the modeled TEC at different latitudes, the results are presented in the box and whiskers plot in Figure 7 for June and December of 2011 to 2013. The box has lines at the lower
quartile, median (red line), and upper quartile values. Whiskers extend from each end of the box to the adjacent values in the data. Outliers beyond the whiskers are displayed using the '+' sign.

To analyse the TEC variations at the grid point $40°$S and $40°$N, for $15°$E each in both hemispheres during June and December (at left, a-d), we compare the observed TEC (O) with the modeled TEC simulated using the SOLAR2000 (S) and the EUVAC (E) flux model for different years. The F10.7 index is used as the primary solar input to calculate the spectra in the model. The
boxplots have been generated using the daily data of June and December, respectively. The right panels show the differences between observed and modeled TEC at different corresponding locations and months (e-h).

The median of modeled TEC using the SOLAR2000 flux model overestimates the observed TEC by about 10 TECU, 11 TECU, and 7 TECU during June 2011, 2012, and 2013, respectively, at $40°$S as shown in Figure 7(a,e). A slightly smaller overestimation can be seen using the EUVAC flux model with a difference of less than about 5 TECU during 2011 and 2013 ,
6 TECU during 2012. Hence both models generally show overestimation of TEC at this latitude and month.

Figure 7(b,f) shows the TEC plot and difference box plot at $40°$N/$15°$E during June. At this grid point, the observed TEC values are high compared to the Southern Hemispheric grid point. The observed TEC is quite comparable with the modeled TEC simulated using SOLAR2000 during 2011 and 2013. However, it shows an overestimation by 2 TECU during 2012. In comparison to SOLAR2000 simulated TEC, the EUVAC model based TEC simulation shows an underestimation of about 5-10
TECU. The modeled TEC using the SOLAR2000 flux model is higher than the one simulated using the EUVAC model. A good agreement between the modeled and observed TEC can be seen at the Southern and Northern hemispheric grid points (Figure 7(e-f)), where the bias is less than 10 TECU. The analysis for December is shown in Figure 7(c-d). The difference plot (Figure 7(g-h)) shows a different behavior than in June. The modeled TEC simulated using the SOLAR2000 is in agreement during December over $40°$S, but the modeled TEC simulated using the EUVAC underestimates the observations by about 10 TECU.

Over the grid point $40°$N, $15°$E, both flux models result in an overestimation, and the SOLAR2000 flux model produces maximum bias during 2011 and 2013, with about 40 TECU, and 20 TECU during 2012. The modeled TEC simulated using the EUVAC model shows an overestimation of about 10 TECU.

The overall difference between the model and observations is larger during December as compared to June. The discrepancy observed in the CTIPe results are possibly due to the various reasons mentioned in the previous section.





Figure 8(a-b) shows the boxplots of TEC for January, June and December during 2013. Here the CTIPe model run used the modified F10.7A index as solar input to calculate the spectra in solar flux models. We choose this period to consider different ionizing radiations. Here the difference plots Figure 8(c-d) show bias during January, June, and December at 40°S/15°E and 40°N/15°E.

At 40°S/15°E, the modeled TEC simulated using the SOLAR2000 flux model is overestimating TEC during January and

December, and underestimating TEC during June by about 5 TECU. The modeled TEC simulated using the EUVAC model shows quite different behavior. It shows overestimation during January and June, but underestimation during December.

In comparison to the Southern hemispheric grid point, the TEC over 40°N /15°E simulated using the SOLAR2000 shows overestimation of TEC and maximum bias during January by about 25 TECU. In the case of the EUVAC model, it shows underestimation during January compared to observed TEC. During June and December, the modeled TEC simulated using

EUVAC shows overestimation with respect to the observed TEC.

Here it is interesting to note that the Southern Hemispheric grid point shows good agreement compared to the Northern Hemisphere. During January, the SOLAR2000 model overestimated TEC by about 20 TECU, while the EUVAC model over-estimated TEC by 5 TECU at 40°N/15°E. The observed TEC shows seasonal variations, while the model is not able to capture seasonal behaviour.

We performed a similar comparison using F10.7A (average of previous 81 days averages with previous day value) as solar input proxy in the solar flux models (not shown). The results show a similar bias as the one presented in Figure 8. The flux values provided by EUVAC are smaller than SOLAR2000 results in the photoionization processes and results in a decrease in TEC.

The large bias observed in the physics-based model is mainly due to the solar EUV flux input and grid resolution. The model

needs further improvement regarding the input of solar flux. Klipp et al. (2019) compared the IGS TEC with the modeled TEC using different flux models (EUVAC and SOLAR2000) over Central and South American regions. They showed different behaviour of empirical models during different solar activity conditions.

## 4  Summary

We presented a climatological analysis of GNSS observed and CTIPe model simulated TEC during three years, 2011 to 2013,

of the 24th solar cycle, to investigate and compare modeled TEC with the observed ones, the ionospheric delay, periodicity estimation, and relation of TEC with the solar proxy. Our results show a distinct low and mid-latitudes TEC response at a longitude of 15°E.

The main results of this study can be summarized as follows:

• The periodicity estimations over the low, mid, and high latitudes show that 16-32 d periodicity was dominant during 2012.

As compared to the periodicity observed in model simulated TEC, the 64-128 d periodicity was missing in the observations over all considered latitudes.





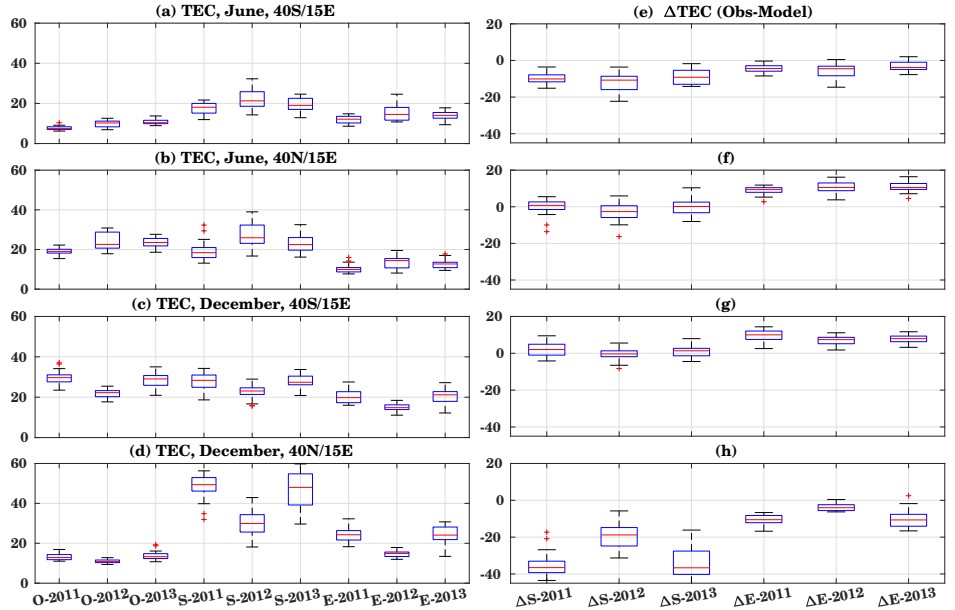

**Figure 7.** Boxplots based on daily TEC during June and December 2011-2013 for $40°$S and $40°$N. The months and location are mentioned in the figure titles. Here O, S, and E represent Observed, CTIPe-SOLAR2000 flux model, CTIPe-EUVAC flux model TEC, respectively. The left panel shows the boxplots for the difference between observed TEC with the model simulated TEC using different flux models. Data points beyond the whiskers are displayed using the '+' sign.

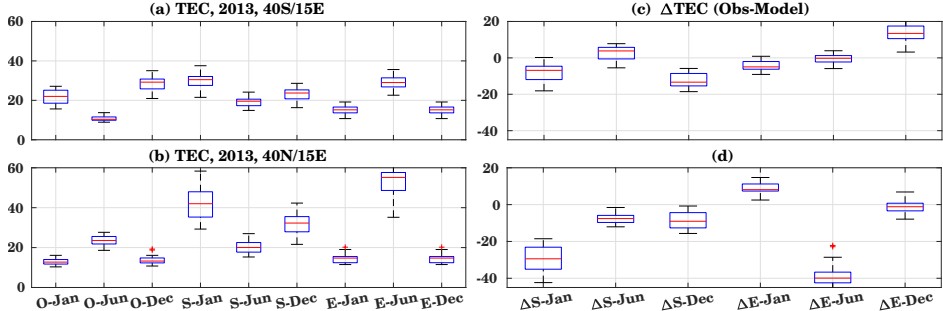

**Figure 8.** Boxplots of observed daily TEC and model simulated TEC using F10.7A as solar input for $40°$S and $40°$N during January, June, and December months for the year 2013.



- While comparing TEC against the F10.7 index, the correlation is higher in 2011 and 2013 over the Southern Hemisphere as compared to the Northern Hemisphere, i.e., there is a hemispheric asymmetry. A similar characteristic has been observed by (Romero-Hernandez et al., 2018). The lowest correlation is observed during 2012.

- The ionospheric delay has been investigated using the modeled and observed TEC against the solar EUV flux. The ionospheric delay estimated using model simulated TEC is in good agreement with the delay estimated for observed TEC. An average delay for the observed (modeled) TEC is about 17 (16) hours. The study confirms the model capabilities to reproduce the delayed ionospheric response against the solar EUV flux. These results are in close agreement with Schmölter et al. (2020).

- The average difference between the Northern and Southern Hemispheric delay estimated for observed (modeled) TEC is
about 1 (2) hours. The average delay is higher in the Northern Hemisphere as compared to the Southern Hemisphere.

- Furthermore, the observed TEC is compared with the modeled TEC simulated using the SOLAR2000 and EUVAC flux models within CTIPe at the Northern and Southern Hemispheric grid points. The analysis indicates that TEC simulated using the SOLAR2000 flux model overestimates the observed TEC, which is not the case when using the EUVAC flux model. The large bias observed in the physics-based model is mainly due to the solar EUV flux input and grid resolution. Our results show
that the model needs further improvement in respect to the solar flux input to further reduce the presented deviation to TEC measurements.

*Data availability.* IGS TEC maps have been provided by NASA through ftp://cddis.gsfc.nasa.gov/gnss/products/ionex (CDDIS, 2018). SDO-EVE data have been provided by the Laboratory for Atmospheric and Space Physics (LASP) through http://lasp.colorado.edu/eve/data_access/evewebdata (LASP, 2018). Daily F10.7 index can be downloaded from
http://lasp.colorado.edu/lisird/data/noaa_radio_flux/ (LASP, 2018).

*Author contributions.* RV together with CJ and MC performed the CTIPe model simulations. RV drafted the first version of the manuscript. ES, CJ, and JB actively contributed to the analysis. All authors discussed the results and contributed to the final version of the manuscript.

*Competing interests.* Christoph Jacobi is one of the Editors-in-Chief of Annales Geophysicae. The authors declare that they have no conflict of interest.

*Acknowledgements.* We acknowledge NASA for providing the IGS TEC data, through ftp://cddis.gsfc.nasa.gov/gnss/products/ionex/ (CDDIS, 2018). SDO-EVE data and daily F10.7 index can be downloaded have been provided by the Laboratory for Atmospheric and Space Physics (LASP, 2018). The study has been supported by Deutsche Forschungsgemeinschaft (DFG) through grants No. BE 5789/2-1 and JA 836/33-1.

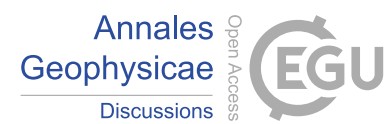

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
