# Peer review of "Ionospheric Response to Solar EUV Radiation Variations: Comparison based on CTIPe Model Simulations and Satellite Measurements"

_Annales Geophysicae, 2020_

## Referee Comment (RC1) · Gerhard Schmidtke (Referee) · 8 Jan 2021

The manuscript focuses on examining the delay time in Total Electron Content (TEC) associated with solar activity as investigated from 70oS to 70oN latitude along the 15oE longitude. Based on the data from the International GNSS Service (IGS) and the Coupled Thermosphere Ionosphere Plasmasphere Electrodynamics (CTIPe) model, changes in TEC data are correlated with solar data relating to changes in the spectral range of the extreme ultraviolet (EUV). The period from years 2011 to 2013 is well chosen because precise data on the Solar Spectral Irradiance (SSI) is available and the EUV variability is pronounced at the first maximum of solar activity during the 24th

solar cycle. The comparison of TEC data changes with EUV data, the SOLAR2000 and EUVAC flux10 models and the solar radio flux index F10.7 leads to a more precise accuracy of delay times from EUV to TEC changes and to improvements in the physics-based Coupled Thermosphere Ionosphere Plasmasphere Electrodynamics (CTIPe) model. In this section, different degrees of correlation with TEC data are clearly explained using the simulated or modeled or measured energy input into the CTIPs model. Taking these results into account, the ionospheric delay time is estimated for the various sources of EUV or EUV-simulated data at different states of solar EUV activity. The EUV-SDO data provide the most reliable values for the TEC time delay. To further investigate the estimated dalay time of 16 houres for the modeled TEC and 17 hours for the observed TEC, the different delay times in the northern and southern hemisphere and related issues to improve the CTIPc, the need for the availability of continous SSI-EUV flux data is clearly expressed. Investigating the correlation between TEC and SSI-EUV is difficult due to the 'spontaneous' occurrence of active sunspot regions on different regions of the solar disk. Could it be helpful to select periods of distinct high EUV activity changes, as from June to December 2013, in order to derive even more preccise delay times? If longer periods are selected, the periodocity is a mixture of lower and higher solar activity. Then the appearance of sunspots at different locations on the solar disk shifts the maximum EUV emissions in relation to coherence with one another, for which the correlation is expected to decrease. An explanation of this problem would be helpful for the reader to interpret the results. Conclusion: The manuscript is clearly structured and well written. It contributes good results on the TEC delay times for the selected geografic region from 70oS to 70oN latitude along the 15oE longitude during the period from 2011 to 2013. If possible, an estimate of the expected improvement by considering the aspect of selecting coherent EUV data periods is suggested. The manuscript is strongly recommended for publication.

---

## Referee Comment (RC2) · Anonymous Referee #2 · 10 Jan 2021

The paper reports the time delay of ionospheric TEC responses to solar EUV irradiance using SDO-measured solar EUV flux, GNSS-based TEC observations, and simulated TEC from first-principle ionospheric model CTIPe. The study finds that the average time delay of about 17 and 16 hours for the observed and modeled TEC responses to EUV irradiance, a hemispheric asymmetry in the time delay, as well as the different CTIPe-simulated TECs using two different solar EUV irradiance models. The paper delivers an interesting and inspiring study with a clearly-presented motivation, methodology, and results. The study contributes to the scientific understanding of the ionospheric responses to solar irradiance and can guide the solar irradiance specification in ionospheric models. I only have some minor comments listed below:

[Figure]

1. Line 69, "ionospheric composition": ionospheric electron density (or ion density) is perhaps more precise here? There are plenty of neutral species in the ionosphere as well, whose densities drop with altitude.

2. Lines 74-75: It would be helpful to include a figure showing the GNSS ground receiver locations around 15 degree E, or at least some justification of how many ground receivers near the region were used to produce the TEC maps.

3. Lines 73-74, "moderate solar activity phase": is the interval during solar inclining or declining phase?

4. Line 177, "mid-day (11:00-13:00LT)": is it an average of the TEC values during 11-13LT?

5. Lines 191-192, "The Figure 2(a) shows the two peaks of ionization during the spring 2011, but in autumn the maximum is shifted towards winter, clearly solar driven, and in 2013 there are local minima during equinox." What exactly are the "two peaks" during spring 2011, and what does "maximum is shifted towards winter" mean? These are not clear from Figure 2(a).

6. Lines 193-195: I suppose the "spring", "summer", "winter" refer to the seasons in the Southern Hemisphere? This should be stated in "The bias between the modeled and observed 195 TEC is higher during the spring and summer season."

7. Line 212, are the TEC averages being taken within low, mid, and high latitude bands?

8. Lines 218-219, "the influence of other dynamical processes in the ionosphere (e.g., lower atmospheric forcing) is stronger.": Is there any evidence supporting this statement? The weak 27d periodicity in F10.7 for 2011 and 2013 does not necessarily imply that the other dynamical processes have a stronger impact. Or the authors refer to the 27 d periodicity in TEC instead of F10.7 here? Line 220, "The 27 d period is stronger in the winter season": Southern Hemisphere winter?
9. Line 246, "daily data of 40N and 40S": Are there GNSS ground receivers nearby 40N and 40S, 15E? How accurate is the GIM TEC map at these two locations?

10. Line 250, "solar radiation": perhaps "solar EUV radiation" to be more precise? because F10.7 proxies the EUV irradiance only.

11. Lines 257-260: attributing the unusual behavior for 2012 to the underlying model in the TEC maps is not convincing, since the underlying model of TEC maps should remain unchanged for different years.

12. Lines 289-290, "the ionospheric delay is increasing with increasing solar activity." — Does this refer to the increasing delay from 2011 to 2013 and the solar activity enhances from 2011 and 2013?

13. Lines 300-301, "This negative correlation indicates the effect of local dynamics.": Can you provide more explanation on this?

14. Line 305, "The observed TEC always overestimated the model simulated TEC at all latitudes.": Given the observed TEC is the "truth", it sounds more natural to say that the model simulated TEC underestimate the observed TEC.

15. Line 364, "The large bias observed in the physics-based model is mainly due to the solar EUV flux input and grid resolution.": How grid resolution impact the agreement between simulated and observed TEC? A justification is necessary.

---

## Author Comment (AC2) · 28 Jan 2021

**Reply to Second Reviewer's Comments:**

**Anonymous Referee #2**

The paper reports the time delay of ionospheric TEC responses to solar EUV irradiance using SDO-measured solar EUV flux, GNSS-based TEC observations, and simulated TEC from first-principle ionospheric model CTIPe. The study finds that the average time delay of about 17 and 16 hours for the observed and modeled TEC responses to EUV irradiance, a hemispheric asymmetry in the time delay, as well as the different CTIPe-simulated TECs using two different solar EUV irradiance models. The paper delivers an interesting and inspiring study with a clearly-presented motivation, methodology, and results. The study contributes to the scientific understanding of the ionospheric responses to solar irradiance and can guide the solar irradiance specification in ionospheric models.

Response: We are thankful for the reviewer's comments and suggestions which help us to improve the quality of the manuscript. We will address all the raised points in the revised version of the manuscript.

I only have some minor comments listed below:

1. Line 69, "ionospheric composition": ionospheric electron density (or ion density) is perhaps more precise here? There are plenty of neutral species in the ionosphere as well, whose densities drop with altitude.
Response: We agree with the reviewer's suggestion. We will improve this in the revised text.

2. Lines 74-75: It would be helpful to include a figure showing the GNSS ground receiver locations around 15 degree E, or at least some justification of how many ground receivers near the region were used to produce the TEC maps.
Response: Thank you. We will include this information in the revised version of the manuscript.

3. Lines 73-74, "moderate solar activity phase": is the interval during solar inclining or declining phase?
Response: The selected interval is during solar inclining phase of solar cycle 24. We will add this information in the revised text.

4. Line 177, "mid-day (11:00-13:00LT)": is it an average of the TEC values during 11-13LT?
Response: Yes, this is an average of the TEC values during 11-13 LT. We will clarify this in the revised version.

5. Lines 191-192, "The Figure 2(a) shows the two peaks of ionization during the spring 2011, but in autumn the maximum is shifted towards winter, clearly solar driven, and in 2013 there are local minima during equinox." What exactly are the "two peaks" during spring 2011, and what does "maximum is shifted towards winter" mean? These are not clear from Figure 2(a).
Response: We apologize for this confusion due to typos. Here two peaks of ionization refer to maximum TEC around the equator during the spring 2011. We will clarify and rephrase the paragraph in the revised version.

6. Lines 193-195: I suppose the "spring", "summer", "winter" refer to the seasons in the Southern Hemisphere? This should be stated in "The bias between the modeled and observed TEC is higher during the spring and summer season."

Response: Yes, "spring", "summer", "winter" refer to the seasons in the Southern Hemisphere. We will improve this in the revised text.

7. Line 212, are the TEC averages being taken within low, mid, and high latitude bands?

Response: Yes, the TEC averages being taken. We will clarify this in the revised version.

8. Lines 218-219, "the influence of other dynamical processes in the ionosphere (e.g., lower atmospheric forcing) is stronger. ": Is there any evidence supporting this statement? The weak 27d periodicity in F10.7 for 2011 and 2013 does not necessarily imply that the other dynamical processes have a stronger impact. Or the authors refer to the 27 d periodicity in TEC instead of F10.7 here?

Response: This sentence refers to the 27 d periodicity in TEC. We will rearrange the paragraph to avoid confusion.

Line 220, "The 27 d period is stronger in the winter season": Southern Hemisphere winter?

Response: We have analysed the period with respect to low-, mid-, and high latitudes. Hence we will improve this sentence.

9. Line 246, "daily data of 40N and 40S": Are there GNSS ground receivers nearby 40N and 40S, 15E? How accurate is the GIM TEC map at these two locations?

Response: There are no specific stations at this grid point, but around that grid point (European region) several ground stations are located (https://www.igs.org/stations/). Therefore, the impact of the applied interpolation in the TEC map calculation is expected to be smaller than in other regions. The accuracy of IGS TEC maps is given with 2-8 TECU (Chen et al., 2020). The mean RMS at 40°N is 6.92 TECU and the mean RMS at 40°S is 7.54 TECU for the whole period.

**Reference:** 1. Chen, P., Liu, H., Ma, Y. and Zheng, N.: Accuracy and consistency of different global ionospheric maps released by IGS ionosphere associate analysis centers, Advances in Space Research, 65(1), 163-174, https://doi.org/10.1016/j.asr.2019.09.042, 2020.
2. https://www.igs.org/products/

10. Line 250, "solar radiation": perhaps "solar EUV radiation" to be more precise? because F10.7 proxies the EUV irradiance only.

Response: We will improve this in the revised text.

11. Lines 257-260: attributing the unusual behavior for 2012 to the underlying model in the TEC maps is not convincing, since the underlying model of TEC maps should remain unchanged for different years.

Response: We agree with the reviewer's concern. We will remove this part from the manuscript.

12. Lines 289-290, "the ionospheric delay is increasing with increasing solar activity." Does this refer to the increasing delay from 2011 to 2013 and the solar activity enhances from 2011 and 2013?

Response: Yes, we will improve this in the revised text.

13. Lines 300-301, "This negative correlation indicates the effect of local dynamics.": Can you provide more explanation on this?

Response: The negative correlation between the solar EUV and ionospheric TEC is still not well understood. As the concept is nearly nonphysical and needs further investigation, it suggests the ionosphere's unique and different behaviour from the normal conditions. The negative correlation suggests an unexpected increase in the TEC during low solar activity conditions. This might be possible due to additional heating sources or unknown factors such as the state of the ionosphere and its dominant physical processes. Another more important factor is lower atmospheric forcing, such as gravity or planetary wave. Gravity waves can induce wave and turbulent fluxes of heat and constituents and influence the upper atmosphere's thermal and compositional structures. These sources might lead to change in the ionosphere's local dynamics and contribute to the additional increase and decrease in the electron density irrespective of actual solar activity conditions.

14. Line 305, "The observed TEC always overestimated the model simulated TEC at all latitudes.": Given the observed TEC is the "truth", it sounds more natural to say that the model simulated TEC underestimate the observed TEC.

Response: Thank you for the suggestion. We will replace this sentence in the revised version.

15. Line 364, "The large bias observed in the physics-based model is mainly due to the solar EUV flux input and grid resolution.": How grid resolution impact the agreement between simulated and observed TEC? A justification is necessary.

Response: The model has a $2^o$ resolution in latitude and $18^o$ resolution in longitude, and the observed TEC is available at $2.5^o/5^o$ lat/long resolution. The CTIPe model does not have sufficient good resolution to capture the small scale physics, such as sources of variability from the lower atmosphere, which are not included except in a statistical sense. The model does not directly include the impact of gravity waves and planetary waves originating from the lower atmosphere. Hence due to poor resolution, all the smaller scale physics is not included in the model and which might cause the bias between observed and model-simulated TEC.

Miyoshi et al. (2018) investigated the effects of the horizontal resolution on the electron density distribution using the Ground to topside model of Atmosphere and Ionosphere for Aeronomy (GAIA). They showed that the model simulation with high horizontal resolution of $1^o$ x $1^o$ produces fluctuations in electron density with periods of less than around 2 hours and length scales less than around 1000 km, which are in good agreement with observations and which are not seen in a low resolution ($2.5^o$ x $2.5^o$) simulation.

**Reference:** Miyoshi, Y., Jin, H., Fujiwara, H., and Shinagawa, H.: Numerical study of traveling ionospheric disturbances generated by an upward propagating gravity wave, Journal of Geophysical Research: Space Physics, 123, 2141–2155, https://doi.org/10.1002/2017JA025110, 2018.

---

## Author Response (AR1)

Response: We are thankful for the reviewer's comments and suggestions which help us to improve the quality of the manuscript. We have revised the paper according to the suggestions and comments.

We discussed most of the comments of reviewer#1 in the first response. Here we summarise the important points which we have been included in the revised version.

The manuscript focuses on examining the delay time in Total Electron Content (TEC) associated with solar activity as investigated from 70ºS to 70ºN latitude along the 15ºE longitude. Based on the data from the International GNSS Service (IGS) and the Coupled Thermosphere Ionosphere Plasmasphere Electrodynamics (CTIPe) model, changes in TEC data are correlated with solar data relating to changes in the spectral range of the extreme ultraviolet (EUV). The period from years 2011 to 2013 is well chosen because precise data on the Solar Spectral Irradiance (SSI) is available and the EUV variability is pronounced at the first maximum of solar activity during the 24th solar cycle. The comparison of TEC data changes with EUV data, the SOLAR2000 and EUVAC flux models and the solar radio flux index F10.7 leads to a more precise accuracy of delay times from EUV to TEC changes and to improvements in the physics-based Coupled Thermosphere Ionosphere Plasmasphere Electrodynamics (CTIPe) model. In this section, different degrees of correlation with TEC data are clearly explained using the simulated or modeled or measured energy input into the CTIPs model. Taking these results into account, the ionospheric delay time is estimated for the various sources of EUV or EUV-simulated data at different states of solar EUV activity. The EUV-SDO data provide the most reliable values for the TEC time delay. To further investigate the estimated dalay time of 16 hours for the modeled TEC and 17 hours for the observed TEC, the different delay times in the northern and southern hemisphere and related issues to improve the CTIPc, the need for the availability of continous SSI-EUV flux data is clearly expressed. Investigating the correlation between TEC and SSI-EUV is difficult due to the spontaneous' occurrence of active sunspot regions on different regions of the solar disk.

Could it be helpful to select periods of distinct high EUV activity changes, as from June to December 2013, in order to derive even more precise delay times?

Response: Thank you for the suggestion. This is an approach that we considered earlier in the investigation but instead of a few months we planned to analyze specifically the 27-day solar rotation period (or one really significant). With such a method higher correlations and more precise delay estimations are expected. Here our interest is in estimating both, times of high and low correlation.
In addition, there are still other factors that can play an important role and impacts on the precision of the delay estimation during the suggested period such as seasonal and annual variations.

If longer periods are selected, the periodicity is a mixture of lower and higher solar activity. Then the appearance of sunspots at different locations on the solar disk shifts the maximum EUV emissions in relation to coherence with one another, for which the correlation is expected to decrease. An explanation of this problem would be helpful for the reader to interpret the results.

Response: We agree with the reviewer's suggestion. We have included this explanation in the revised version of the manuscript. Pages: 11-12, Line: 278-285

'The cross correlation was applied on independent monthly datasets from 2011 to 2013, as the maximum correlation is expected during the solar rotation period. In the case of longer periods, the periodicity is a mixture of lower and higher solar activity. Then the appearance of sunspots at different locations on the solar disk shifts the maximum EUV emissions in relation to coherence with one another, for which the correlation is expected to decrease. Even shorter periods can result in lower correlations due to the reduced sampling size, i.e. stronger impact of smaller deviations, as well. Similar results have been shown by Vaishnav et al. (2019). They studied correlation analysis between TEC and multiple solar proxies for different time periods. Their study revealed that the correlation is lower during shorter and longer periods. Better correlations are only expected during the solar rotation period.'

**Reference:** Vaishnav, R., Jacobi, C., and Berdermann, J.: Long-term trends in the ionospheric response to solar extreme-ultraviolet variations, Ann. Geophys., 37, 1141–1159, https://doi.org/10.5194/angeo-37-1141-2019, 2019.

Conclusion: The manuscript is clearly structured and well written. It contributes good results on the TEC delay times for the selected geographic region from 70ºS to 70ºN latitude along the 15ºE longitude during the period from 2011 to 2013. If possible, an estimate of the expected improvement by considering the aspect of selecting coherent EUV data periods is suggested. The manuscript is strongly recommended for publication.

Response: Thank you for reviewing our manuscript.

**Reply to Second Reviewer's Comments:**

**Anonymous Referee #2**

The paper reports the time delay of ionospheric TEC responses to solar EUV irradiance using SDO-measured solar EUV flux, GNSS-based TEC observations, and simulated TEC from first-principle ionospheric model CTIPe. The study finds that the average time delay of about 17 and 16 hours for the observed and modeled TEC responses to EUV irradiance, a hemispheric asymmetry in the time delay, as well as the different CTIPe-simulated TECs using two different solar EUV irradiance models. The paper delivers an interesting and inspiring study with a clearly-presented motivation, methodology, and results. The study contributes to the scientific understanding of the ionospheric responses to solar irradiance and can guide the solar irradiance specification in ionospheric models.

Response: We are thankful for the reviewer's comments and suggestions which help us to improve the quality of the manuscript. We have revised the paper according to the suggestions and comments.

We discussed most of the comments of reviewer#2 in the first response. Here we summarise the important points which we have been included in the revised version.

I only have some minor comments listed below:

1. Line 69, "ionospheric composition": ionospheric electron density (or ion density) is perhaps more precise here? There are plenty of neutral species in the ionosphere as well, whose densities drop with altitude.
Response:  We have improved this in the revised text. Page: 3, Line: 70

2. Lines 74-75: It would be helpful to include a figure showing the GNSS ground receiver locations around 15 degree E, or at least some justification of how many ground receivers near the region were used to produce the TEC maps.
Response: We have included a figure (Figure 1) showing GNSS stations around 15 E. Page: 7

3. Lines 73-74, "moderate solar activity phase": is the interval during solar inclining or declining phase?
Response: The selected interval is during the solar inclining phase of solar cycle 24. We have added this information in the revised text. Page: 3, Line: 75

4. Line 177, "mid-day (11:00-13:00LT)": is it an average of the TEC values during 11-13LT?
Response: Yes, this is an average of the TEC values during 11-13 LT. We have added this in the revised version. Page: 8, Line: 180

5. Lines 191-192, "The Figure 2(a) shows the two peaks of ionization during the spring 2011, but in autumn the maximum is shifted towards winter, clearly solar driven, and in 2013 there are local minima during equinox." What exactly are the "two peaks" during spring 2011, and what does "maximum is shifted towards winter" mean? These are not clear from Figure 2(a).
Response: We have rephrased the paragraph in the revised version. Page: 8, Line: 195-197

'Figure 3(a) shows maximum TEC around the equator during the December solstice, and a minimum TEC is observed during the June solstice of 2011 coincides with the minimum solar EUV flux. There are local minima during equinoxes in 2013.'

6. Lines 193-195: I suppose the "spring", "summer", "winter" refer to the seasons in the Southern Hemisphere? This should be stated in "The bias between the modeled and observed TEC is higher during the spring and summer season."

Response: Yes, "spring", "summer", "winter" refer to the seasons in the Southern Hemisphere. We have improved this in the revised text. Page: 8, Line: 200

7. Line 212, are the TEC averages being taken within low, mid, and high latitude bands?

Response: Yes, the TEC averages were taken. We have included this in the revised version. Page: 9, Line: 218

8. Lines 218-219, "the influence of other dynamical processes in the ionosphere (e.g., lower atmospheric forcing) is stronger. ": Is there any evidence supporting this statement? The weak 27d periodicity in F10.7 for 2011 and 2013 does not necessarily imply that the other dynamical processes have a stronger impact. Or the authors refer to the 27 d periodicity in TEC instead of F10.7 here?

Response: This sentence refers to the 27 d periodicity in TEC. We have rearranged the paragraph to avoid confusion. Page: 9, Line: 222-228

'The CWT of modeled TEC shows the dominant 16-32 d oscillations during 2012. This is, however, not the case during 2011 and 2013. During these periods, the influence of other dynamical processes in the ionosphere (e.g., lower atmospheric forcing) is stronger.

During these years, very weak 27 d periodicity is observed. The 27 d period is stronger during December and January. Pancheva et al. (1991) showed that the 27 d variation in the lower ionosphere (D region) is often caused by dynamical forcing (planetary waves), particularly in the winter season under low solar activity. A similar 16-32 days periodicity is observed in the F10.7 index. It is well known that the 27 d periodicity is one of the major and dominant modes of variations in the solar proxies.'

Line 220, "The 27 d period is stronger in the winter season": Southern Hemisphere winter?

Response: We have analysed the period with respect to low-, mid-, and high latitudes. Hence we have improve this sentence. Page: 9, Line: 224

'The 27 d period is stronger during December and January.'

9. Line 246, "daily data of 40N and 40S": Are there GNSS ground receivers nearby 40N and 40S, 15E? How accurate is the GIM TEC map at these two locations?

Response: There are no specific stations at this grid point, but around that grid point (European region) several ground stations are located (https://www.igs.org/stations/). Therefore, the impact of the applied interpolation in the TEC map calculation is expected to be smaller than in other regions. The accuracy of IGS TEC maps is given with 2-8 TECU (Chen et al., 2020). The mean RMS at 40°N is 6.92 TECU and the mean RMS at 40°S is 7.54 TECU for the whole period.

10. Line 250, "solar radiation": perhaps "solar EUV radiation" to be more precise? because F10.7 proxies the EUV irradiance only.

Response: We have improved this in the revised text. Page: 11, Line: 257

11. Lines 257-260: attributing the unusual behavior for 2012 to the underlying model in the TEC maps is not convincing, since the underlying model of TEC maps should remain unchanged for different years.
Response: We have removed this part from the manuscript.

12. Lines 289-290, "the ionospheric delay is increasing with increasing solar activity." Does this refer to the increasing delay from 2011 to 2013 and the solar activity enhances from 2011 and 2013?
Response: Yes, we have improved this in the revised text. Page: 12, Line: 303

13. Lines 300-301, "This negative correlation indicates the effect of local dynamics.": Can you provide more explanation on this?
Response: We have added a brief description in the revised version of manuscript. Pages: 12-13, Line: 313-318

'This negative correlation might be possible due to additional heating sources or unknown factors such as the state of the ionosphere and its dominant physical processes. Another more important factor is lower atmospheric forcing, such as gravity or planetary wave. Gravity waves can influence the upper atmosphere's thermal and compositional structures. These sources might lead to changes in the ionosphere's local dynamics and contribute to additional increase and decrease in the electron density irrespective of actual solar activity conditions. '

14. Line 305, "The observed TEC always overestimated the model simulated TEC at all latitudes.": Given the observed TEC is the "truth", it sounds more natural to say that the model simulated TEC underestimate the observed TEC.
Response: We have replaced this sentence in the revised version. Page: 13, Line: 322

15. Line 364, "The large bias observed in the physics-based model is mainly due to the solar EUV flux input and grid resolution.": How grid resolution impact the agreement between simulated and observed TEC? A justification is necessary.
Response:  We have included the justification in the revised manuscript paragraph. Page: 16, Line: 386-390

'Miyoshi et al. (2018) investigated the effects of the horizontal resolution on the electron density distribution using the GAIA model. They showed that fluctuations produced in model simulated electron density with periods of less than about 2 hours and length scales less than about 1000 km with a high horizontal resolution of 1º x 1º, which are in good agreement with observations. These fluctuations are not seen in a low resolution (2.5º x 2.5º) simulation.
Hence, the model resolution is an important factor for the large bias between observations and model simulations.'